# Products Derived from *Buchenavia tetraphylla* Leaves Have In Vitro Antioxidant Activity and Protect *Tenebrio molitor* Larvae against *Escherichia coli*-Induced Injury

**DOI:** 10.3390/ph13030046

**Published:** 2020-03-16

**Authors:** Tiago Fonseca Silva, José Robson Neves Cavalcanti Filho, Mariana Mirelle Lima Barreto Fonsêca, Natalia Medeiros dos Santos, Ana Carolina Barbosa da Silva, Adrielle Zagmignan, Afonso Gomes Abreu, Ana Paula Sant’Anna da Silva, Vera Lúcia de Menezes Lima, Nicácio Henrique da Silva, Lívia Macedo Dutra, Jackson Roberto Guedes da Silva Almeida, Márcia Vanusa da Silva, Maria Tereza dos Santos Correia, Luís Cláudio Nascimento da Silva

**Affiliations:** 1Departamento de Bioquímica, Universidade Federal de Pernambuco, Recife/PE 50670-901, Brazil; tiagotfs89@gmail.com (T.F.S.); robsonncavalcanti@gmail.com (J.R.N.C.F.); marianamirelle@gmail.com (M.M.L.B.F.); natmmedeiros@gmail.com (N.M.d.S.); annapsantanna@hotmail.com (A.P.S.d.S.); lima.vera.ufpe@gmail.com (V.L.d.M.L.); nhsilva@uol.com.br (N.H.d.S.); marciavanusa@yahoo.com.br (M.V.d.S.); mtscorreia@gmail.com (M.T.d.S.C.); 2Faculdade de Odontologia, Universidade de Pernambuco, Camaragibe/PE 54756-220, Brazil; 3Curso de Farmácia, Faculdade Pernambucana de Saúde, Recife/PE 51150-000, Brazil; 4Curso de Biomedicina, Universidade Ceuma, São Luís/MA 65075-120, Brazil; carolinabsilva07@gmail.com; 5Curso de Nutrição, Universidade Ceuma, São Luís/MA 65075-120, Brazil; adrielle.zagmignan@ceuma.br; 6Programas de Pós-graduação, Universidade Ceuma, São Luís/MA 65075-120, Brazil; afonso.abreu@ceuma.br; 7Núcleo de Estudos e Pesquisas de Plantas Medicinais, Universidade Federal do Vale do São Francisco, Petrolina/PE 56304-205, Brazil; liviamdutra@gmail.com (L.M.D.); jackson.guedes@univasf.edu.br (J.R.G.d.S.A.)

**Keywords:** oxidative stress, natural products, medicinal plants, anti-infective agents, alternative infection models

## Abstract

The relevance of oxidative stress in the pathogenesis of several diseases (including inflammatory disorders) has traditionally led to the search for new sources of antioxidant compounds. In this work, we report the selection of fractions with high antioxidant action from *B. tetraphylla* (BT) leaf extracts. *In vitro* methods (DPPH and ABTS assays; determination of phenolic and flavonoid contents) were used to select products derived from *B. tetraphylla* with high antioxidant action. Then, the samples with the highest potentials were evaluated in a model of injury based on the inoculation of a lethal dose of heat-inactivated *Escherichia coli* in *Tenebrio molitor* larvae. Due to its higher antioxidant properties, the methanolic extract (BTME) was chosen to be fractionated using Sephadex LH-20 column-based chromatography. Two fractions from BTME (BTFC and BTFD) were the most active fractions. Pre-treatment with these fractions protected larvae of *T. molitor* from the stress induced by inoculation of heat-inactivated *E. coli*. Similarly, BTFC and BTFD increased the lifespan of larvae infected with a lethal dose of enteroaggregative *E. coli* 042. NMR data indicated the presence of aliphatic compounds (terpenes, fatty acids, carbohydrates) and aromatic compounds (phenolic compounds). These findings suggested that products derived from *B. tetraphylla* leaves are promising candidates for the development of antioxidant and anti-infective agents able to treat oxidative-related dysfunctions.

## 1. Introduction

A substantial amount of evidence has indicated the key role of free radicals and reactive oxygen species (ROS) in the etiology of degenerative pathologies associated with aging (Parkinson’s and Alzheimer’s diseases), cancer, cardiovascular diseases, and diabetes [1,2,3,4,5,6,7]. Free radicals are highly reactive molecules characterized by having unpaired electrons in the last valence layer, thus becoming potent oxidizing agents [8,9]. These entities are produced as a result of normal cellular metabolism and play an important role in cell function and signaling; however, they can also damage important macromolecules (DNA, proteins, and lipids), thereby impairing cellular functions and leading to cell death [2,3,8].

Due to the reactivity of free radicals, organisms have developed an efficient antioxidant defense system formed by enzymes (such as superoxide dismutase and catalase) and proteins (glutathione reductase, thioredoxin) [8,9]. However, in many situations, this system cannot cope with the overproduction of reactive species, generating a so-called oxidative stress state, which is related to the clinical manifestations described above [1]. An alternative way of combatting the damage caused by free radicals is the use of exogenous substances collectively called antioxidants [4,8,10].

The antioxidants (natural or synthetic) can act through different mechanisms in the organism, such as: (i) direct neutralization of free radicals; (ii) expression of molecules from the host antioxidant defense systems; (iii) inhibition of oxidant enzymes (leading to reduction of free radicals’ generation/propagation) [10,11,12,13]. These compounds can also be used in the food industry to maintain the physical-chemical quality of fruits, meat, and other foods [14,15,16]. Due to the side effects of synthetic antioxidants, those from natural sources are preferred; this context leads to a constant search for plant-derived compounds with this property [4,10,11,12,14,15,16,17,18].

In addition, antioxidant compounds have been related to the therapeutic properties of the species considered medicinal. In fact, substances with antioxidant actions have been detected in different products derived from plants (juices, teas, extracts, infusions) used in the treatment and prevention of diseases [10,11,12,14,17]. However, the antioxidant potential of some medicinal plants is still unexploited; and the neotropical species called *Buchenavia tetraphylla* (Aubl.) RA Howard (synonymy *Buchenavia capitata*; Combretoideae family) is a good example. This plant is distributed from Cuba (Central America) to southeastern Brazil (South America). *B. tetraphylla* is popularly known as “tanimbuca” in Brazil, where it has ethnomedicinal importance for communities in the northeast region [19]. It is also known to have a broad spectrum of antimicrobial activity, inhibiting bacteria, fungi, and virus [20,21,22]. Buchenavianine and two derivatives (O-demethylbuchenavianine, N,O-bis-(demethyl)buchenavianine) have been isolated from *B. tetraphylla*. These compounds are classified as flavoalkaloids with a piperidine moiety at carbon 8 [20,23].

The antioxidant potential of plant products has been traditionally characterized by in vitro methods; however, the biological relevance of these tests has been contested by several works [10,24,25]. In general, in vitro methods have been limited to sample prospection and compound isolation. Thus, the need to employ cell-based methods and in vivo models for a better understanding of the pharmacological action of a candidate as an antioxidant agent is evident [10]. In this work, in vitro methods were used to select products derived from *B. tetraphylla* with antioxidant action. Then, the samples with the highest potentials were evaluated in alternative models of stress based on *Tenebrio molitor* larvae inoculated with *Escherichia coli*.

## 2. Results

### 2.1. Comparison of the Antioxidant Activity of Extracts Obtained from B. tetraphylla Leaves

Initially, the phenolic and flavonoid contents were compared in different extracts of *B. tetraphylla* (BTHE: hexane extract; BTCE: chloroform extract; BTEE: ethyl acetate extract; BTME: methanolic extract) (Table 1). Among the extracts, BTME presented higher concentrations of both classes of compounds with values of 123.03 ± 1.51 mg of gallic acid equivalent (GAE) per mg of dry extract (mg GAE/mg) and 108.90 ± 0.07 mg of quercetin equivalent (QE) per mg of dry extract (mg QE/mg) (*p* < 0.05). A Pearson coefficient of 0.71 was found between the phenolic and flavonoid contents, indicating a strong correlation.

The antioxidant potential of the extracts and fractions was evaluated using the DPPH (2,2-diphenyl-1-picrylhydrazyl radical) and ABTS ((2,2-azino-bis (3-ethylbenzo-thiazoline-6-sulfonic acid) radical) methods. In the DPPH assay, the highest in vitro antioxidant activity was observed for BTME with an EC_50_ (half maximal effective concentration) of 79.04 μg/mL (Table 1) and higher scavenging action in almost all tested concentrations in relation to other extracts (*p* < 0.001). The EC_50_ found for Trolox was 44.10 μg/mL (positive control). Similarly, in the ABTS assay, a greater scavenging action was observed for BTME (approximately 100%; *p* < 0.05), followed by BTEE (approximately 50%) (Figure 1A). Strong correlations were also found between the levels of phenolic compounds and the scavenger actions towards DPPH (0.96) and ABTS radicals (0.89). The flavonoid contents were moderately correlated with DPPH scavenging (0.68) and strongly correlated with ABTS scavenging (0.95). The results for both antioxidant assays were strongly related (0.85).

### 2.2. Comparison of Phenolic and Flavonoid Content and Antioxidant Activity of Fractions Obtained from Methanolic Extract of B. tetraphylla Leaves

Since a higher level of antioxidant activity was observed in BTME, it was submitted to fractionation using Sephadex LH-20 column chromatography. A total of 9 non-repetitive fractions were obtained (BTFA to BTFI). Among them, the highest phenolic content was detected in BTFD (168.99 ± 2.22 μg GAE/μg), followed by BTFC (156.02 ± 4.51 μg GAE/μg), BTFG (127.62 ± 19.11 μg GAE/μg), and BTFI (110.15 ± 0.78 μg GAE/μg) (Table 2). Almost the same pattern was observed for the flavonoid content; in this case, BTFC had the highest values (68.26 ± 2.87 μg QE/μg) (*p* < 0.0001), followed by BTFD (56.01 ± 5.54 μg QE/μg), BTFG (45.27 ± 4.13 μg QE/μg), and BTFI (39.29 ± 2.89 μg QE/μg) (Table 2). A strong correlation (Pearson coefficient of 0.88) was observed among the concentration of phenolic and flavonoid compounds in the fractions obtained from BTME. Following, the antioxidant action of each fraction was investigated, and BTFC showed the highest activity against the DPPH radical (EC_50_: 50.41 μg/mL), followed by BTFD (EC_50_: 237.7641 μg/mL), BTFG (EC_50_: 294.38 μg/mL), and BTFI (EC_50_: 376.25 μg/mL). On the other hand, the fractions BTFB, BTFC, BTFD, BTFF, and BTFI scavenged approximately 80% of the ABTS radical (*p* < 0.05), and no statistical differences were observed between them (Figure 1B).

### 2.3. Evaluation of the Hemolytic Effects of Extracts and Fractions from Buchenavia tetraphylla Leaves

Further, the hemolytic potential of each extract or fraction was evaluated using human erythrocytes (Figure 2 and Figure 3). BTHE, BTCE, and BTEE induced toxic effects when tested at the highest concentrations (500 μg/mL and 1000 μg/mL) (Figure 2A–C). In contrast, it was observed that the BTME and its fractions did not induce significant hemolytic activity, even at the highest tested concentrations (Figure 2D and Figure 3).

### 2.4. Effects of Extracts and Fractions from Buchenavia tetraphylla Leaves on the Survival of Tenebrio molitor larvae Submitted to Stress Induced by Heat-Killed Escherichia coli

Based on the results presented in the above activities, we decided to evaluate the effects of the methanolic extract and its most active fractions (BTFC and BTFD) in a model of stress induced by heat-killed *E. coli* OP50 in *T. molitor* larvae. The heat treatment was used to ensure that larval death was not induced by bacterial growth; it was caused by the components present in the bacteria, such as lipopolysaccharide (LPS). Additionally, we used the nonpathogenic *E. coli* OP50 strain [26]. In this sense, at this point, we did not evaluate the antimicrobial action of the extract/fractions, but the ability of these to inhibit stress pathways induced by the presence of the bacteria. First, we evaluated the effects of several concentrations (measured at OD_600_ (optical density at 600 nm)) of heat-killed *E. coli* OP50 (data not shown). The best results were obtained with the suspension at an OD600 of 0.7; this dose induced the death of 50% of the larvae after 15 h and of 90% after 30 h. These larvae presented typical myelination points related to stress induction in this organism. The pre-treatment with BTME at 10 mg/kg was able to inhibit animal death, with survival rates of 100%, 70%, and 50% after 15 h, 30 h, and 60 h of infection, respectively. The concentration of 20 mg/kg showed no significant protective action (Figure 4A).

The results obtained with BTFD were even more significant. The group treated with 20 mg/kg exhibited viability rates of 90% and 80% after 45 h and 60 h, respectively. Similar results were obtained with BTFD at 10 mg/kg, where viability remained at 80% after 45 h and 70% after 60 h (Figure 4C). Regarding BTFC, the best protective action was observed for the dose of 10 mg/kg. In this case, survival rates remained at 70% up to 45 h and ended at 60% (60 h). In the case of the 20 mg/kg dose, after 15 h, 80% of the larvae remained viable, and this rate progressively decreased to 50% (30 h), 40% (45 h), and reached 30% after 60 h (Figure 4B).

### 2.5. Effects of Fractions from Buchenavia tetraphylla Leaves in the Lifespan of Tenebrio molitor larvae Infected by Enteroaggregative Escherichia coli

We also evaluated the efficacy of BTFC and BTFD in a model of infection provoked by Enteroaggregative *E. coli* 042. The larvae were treated with each fraction 2 h after the infection. *E. coli* 042 killed most of the larvae in less than 24 h (median survival of one day). In contrast, the group treated with *B. tetraphylla* had median survivals of two days (Figure 5). It is important to highlight that these fractions did not have antimicrobial activity towards *E. coli*.

### 2.6. Nuclear Magnetic Resonance Analysis of Buchenavia tetraphylla Leaves

The ^1^H NMR analysis and two-dimensional experiments revealed similar profiles for BTFC and BTFD. The intense overlapping made difficult the identification of metabolites; however, it was possible to suggest the presence of some classes of compounds. The fractions showed signals in spectral regions related to aliphatic and aromatic compounds (Figure 6). The signals at δ 0.50–5.90 ppm attributed to an aliphatic compound (such as terpenoids, fatty acids, carbohydrates) were more expressive in comparison to the signals at δ 6.00–8.50 ppm related to aromatic compounds (such as phenolic compounds). This was observed mainly in the BTFD fraction.

## 3. Discussion

The role of oxidative stress in the development of severe clinical conditions has promoted the search for new antioxidant agents from natural products [1,2,10,24,25]. Herein, we report the selection of high antioxidant fractions from *B. tetraphylla* leaf extracts using in vitro and in vivo approaches. Products derived from this plant have been shown as potential candidates for the development of new antimicrobial agents [20,21,22]; however, their antioxidant actions have not been properly addressed. The in vitro results obtained in this study showed that the methanolic extract (BTME) had a higher antioxidant activity, and this effect was correlated with its higher phenolic and flavonoid content. Among the fractions obtained from this extract, the best potentials were found for BTFC and BTFD (these fractions also exhibited greater levels of phenolic compounds). Furthermore, these agents were not toxic towards human erythrocytes.

These samples were selected to be evaluated in a model of stress induced by heat-killed *E. coli* in *T. molitor* larvae. This insect has been used as a model organism in studies of microbial pathogenesis and drug development (antimicrobial, antivirulence, and immunomodulator agents) [27,28,29]. Several factors have supported the use of this animal. First, *T. molitor* is susceptible to pathogens such as *Candida albicans*, *E. coli*, and *Staphylococcus aureus*, which are able to persist within the infected insect, and cause changes in tissues, hemolymph, and phagocytes [30,31,32]. Second, the immune system of this insect has some known signaling pathways, such as the Toll pathway, the prophenoloxidase cascade, and the autophagy pathway [33,34,35,36].

The antioxidant defense system of *T. molitor* is composed of several antioxidant and detoxifying enzymes such as superoxide dismutases, peroxidases, catalases, as well as tyrosinase, acetylcholinesterase, carboxylesterase, and glutathione S-transferase [37,38,39]. Previous studies have shown that during infection, the *T. molitor* larvae overproduce reactive species in response to the pathogen presence, leading to increased activity of several antioxidants and detoxifying enzymes that are correlated with larvae death [37,38,40].

The ability of *T. molitor* to produce reactive species in response to deleterious stimulus makes this insect a potential model for the study of antioxidant substances. Despite these advantages, no study has exploited *T. molitor* larvae for the screening of plant-derived antioxidant compounds. The protective action of the agent is evidenced by the increased survival of the treated larvae compared to untreated larvae. This approach using larvae could bring more information than those traditionally used for antioxidant prospecting, based on the chemical interaction of compounds and without biological relevance [10,24].

Since *T. molitor* is a multicellular organism, this approach also presents advantages over those that use cells as some insights into the toxicity of the antioxidant agent can also be assessed. Furthermore, the use of this insect offers some advantages in relation to *Caenorhabditis elegans*, the invertebrate organism traditionally used for in vivo antioxidant evaluation [41,42], such as ease of handling and direct inoculation of the compound agent in the larvae. In our in vivo model of stress induced by heat-killed *E. coli* (OP50 strain), BTFD induced higher protective effects than BTME and BTFC. However, in our assays using the live EAEC strain, the fractions had similar results (both increasing the larvae median survival in one day). It is important to highlight that BTFC and BTFD did no display antimicrobial activity *in vitro* (data not shown), suggesting that their protective effects are related to the antioxidant properties. In this sense, we showed the efficacy of two fractions rich in antioxidants to reduce the deleterious effects of *E. coli*-induced injury in *T. molitor* larvae.

Previous works reported the predominant presence of flavonoids and alkaloids in *B. tetraphylla* and other species from the same genus [20,21,43,44]. In this present research, the most bioactive fractions (BTFC and BTFD) showed a similar chemical composition with the presence of aliphatic (terpenes, fatty acids, carbohydrates) and aromatic compounds (phenolic compounds). In general, the pharmacological potentials of some terpenes are associated with their antioxidant action [45,46]. These studies were reviewed by Gonzalez-Burgos and Gomez-Serranillos [45], who highlighted some structural features involved in the antioxidant action of each type of terpene.

Among the classes of compounds detected, phenolic compounds are the most usually related to antioxidant activity. The high antioxidant abilities of phenolic compounds are related to their phenolic hydroxyl groups that can donate hydrogen atom or transfer electrons, resulting in the scavenging of harmful free radicals (such as hydroxyl radicals). The aromatic groups present in the phenolic acids can also delocalize the unpaired electron [47,48]. According to Dai and Mumper [47], the flavonoids are able to perform electron transfer (mainly due to the ortho-dihydroxy structure on the B ring) and electron delocalization (the 2,3-double bond with a 4-oxo function in the C ring, which relocates from the B ring). The authors also highlighted the essential role of 3- and 5-hydroxyl groups with the 4-oxo function in A and C rings and 3-hydroxyl groups.

## 4. Material and Methods

### 4.1. Collection and Extract Preparation

Leaves of *B. tetraphylla* were collected in November 2013, in Catimbau National Park (Pernambuco, Brazil). The samples were processed according to the taxonomic techniques, identified, and deposited in the Herbarium of Agronomic Institute of Pernambuco (voucher: IPA 80349). The leaves of *B. tetraphylla* were subjected to drying at room temperature and then pulverized using a Macsalab mill (Model 200 LAB). This material was stored in a closed, dark container until used.

For extraction, 25 g of the powder were mixed with 100 mL of the first solvent (hexane; for BTHE) on a rotary shaker table (125 rpm) at 25 °C. After 72 h, the sample was filtered, and the extracted liquid was dried in a rotary evaporator (45 rpm) at 50 °C. The remaining leaf residue was further extracted with 100 mL of chloroform (BTCE), and the above procedure was repeated completely, subsequently performed with ethyl acetate (BTEE), and finally, with methanol (BTME).

### 4.2. Fractionation of the Methanolic Extract

The methanolic extract (2 g) was fractionated by chromatography using a column (80 cm × 2.5 cm) incorporated with Sephadex LH-20 (GE Healthcare^®^, Chicago, IL, USA; 60 cm high). The systems of eluents were based on the combination of methanol and ethyl acetate (as shown below). A total of 120 fractions (7 mL each) were obtained and analyzed by fluorescent black light (25W; 127V; Empalux^®^, Curitiba, Brazil) and thin layer chromatography (POLYGRAM^®^ SIL G60/UV_254_; 20 × 20 cm; 0.20 mm; Macherey-Nagel^®^, Düren, Germany). After these procedures, fractions with similar phytochemical profiles were gathered, resulting in 9 different fractions: BTFA (Fractions 1–4), BTFB (Fractions 5–11), BTFC (Fractions 12–18), BTFD (Fractions 19–26), BTFE (Fractions 27–36), BTFF (Fractions 37–54), BTFG (Fractions 55–61), BTFH (Fractions 62–69), and BTFI (Fractions 70–120). The elution systems (methanol:ethyl acetate; *v*/*v*) for obtaining each fraction were: 7:3 for BTFA; 6:4 for BTFB, BTFC, BTFD, BTFE, BTFF; 5:5 for BTFG; 6:4 for BTFH and BTFI.

### 4.3. Total Phenolic Content

Obtaining the total phenolic compounds in crude extracts and fractions was performed using the Folin-Ciocalteu reagent [49]. Samples of each extract/fraction (200 μL at 1000 μg/mL) were mixed with 1.0 mL of Folin-Ciocalteu reagent, and 800 μL of 20% sodium carbonate were added after 3 min. The mixture was incubated at room temperature, protected from light, and allowed to stand for 2 h. The absorbance of the mixture was measured at 765 nm (GeneQuant 1300, GE Healthcare). The total phenolic content was calculated in mg of gallic acid equivalent (GAE) per mg of dry extract using a calibration curve obtained with gallic acid (y = 0.0043x + 0.0153; R^2^ = 0.9932). The results were expressed as mean ± standard deviation calculated from three independent assays performed in triplicate (*n = 3*).

### 4.4. Flavonoid Content

For flavonoid content, 100 μL (at 1000 μg/mL) of each sample were mixed with 100 μL of the reagent solution (2 g of aluminum chloride diluted in 2% ethanol solution). The mixture was incubated at room temperature and protected from light, and after 60 min, the absorbance was measured at 420 nm [50]. The content of flavonoids was calculated in mg of quercetin equivalent (QE) per mg of dry extract using a calibration curve constructed with standard quercetin solution (y = 0.004x + 0.0121; R^2^ = 0.993). The results are expressed as the mean ± standard deviation calculated from three independent assays performed in triplicate (*n = 3*).

### 4.5. DPPH Assay

An aliquot of 250 μL of 1 mM DPPH solution (2,2-diphenyl-1-picrylhydrazyl; Sigma-Aldrich) was added to 40 μL of different sample concentrations (31.25–1000 μg/mL) and homogenized. After 30 min, the absorbance was measured at 517 nm [51]. Trolox was used as the control compound. The DPPH sequestering activity was calculated using the formula below. The results were expressed as the mean ± standard deviation calculated from three independent assays performed in triplicate (*n = 3*).
DPPH scavenging (%) = (Ac − As)/Ac × 100
where: Ac = absorbance control; As = sample absorbance

### 4.6. ABTS Assay

The ABTS (2,2-azino-bis (3-ethylbenzo-thiazoline-6-sulfonic acid)) radical cation was prepared 16 h prior to the assay by mixing 5 mL of the stock solution (7 mM) with 88 µL of the 140 mM potassium persulfate solution. Aliquots (20 μL) of each extract/fraction and 2 mL of the ABTS radical were mixed, and the absorbance of the solutions was monitored at 734 nm after 6, 15, 30, 45, 60, and 120 min, respectively [52]. Gallic acid was used as the positive control. The ABTS scavenging was calculated using the formula below. The results were expressed as the mean ± standard deviation calculated from three independent assays performed in triplicate (*n = 3*).
ABTS scavenging (%) = (Ac − Aa)/Ac × 100
where Ac (control absorbance) and Aa (sample absorbance).

### 4.7. Hemolytic Activity

Blood (5–10 mL) was obtained from healthy, non-smoker volunteers by venipuncture, after signing a free informed consent form. Human erythrocytes from citrated blood were immediately isolated by centrifugation at 1500 rpm for 10 min. After removal of the plasma, the erythrocytes were washed three times with phosphate-buffered saline (PBS; pH 7.4), and then, a suspension of 1% erythrocytes was prepared as the same buffer. Following, an aliquot of 1.1 mL of erythrocyte suspension was mixed with 0.4 mL of each extract/fraction (concentration range: 125 to 1000 μg/mL). The negative control and positive control received 0.4 mL of PBS and Triton X, respectively. After 60 min of incubation at room temperature, the cells were centrifuged, and the supernatant was used to measure the absorbance of hemoglobin released at 540 nm [21]. The hemolytic activity was expressed in relation to the action of Triton X-100 and calculated using the formula below. The results were expressed as the mean ± standard deviation calculated from three independent assays performed in quadruplicate (*n = 4*).
Hemolytic activity (%) = [(Aa − Ab) × 100]/(Ac − Ab)
where: Aa is sample absorbance; Ab is the absorbance of the negative control; and Ac is the absorbance of the positive control.

### 4.8. Toxicity Model Using Heat-Killed E. coli

Larvae of *T. molitor* (~100 mg) were randomly allocated into groups (*n = 10*). After anesthesia and disinfection, 10 μL of the most active samples (methanolic extract or fractions C and D; at 10 mg/kg or 20 mg/kg) were injected in the ventral membrane between the second and third abdominal segments (tail to the head) [29]. One hour after the sample inoculation, the larvae received 10 μL of heat-killed *E. coli* OP50 (optical density at 600 nm: 0.7). The viability of the larvae was evaluated after 15, 30, 45, and 60 h (by evaluation the lack of movement after mechanical stimulus). Larvae inoculated with the microorganism and treated with PBS were used as the negative control; while larvae that received two doses of PBS were the positive control. The experiment was performed in three independent assays.

### 4.9. Infection Model Using Enteroaggregative E. coli

Larvae (*n = 10*) were prepared as described above and infected with 10 μL of enteroaggregative *E. coli* (EAEC) 042 (optical density at 600 nm: 0.1). After two hours, each animal received 10 μL of BTFC or BTFD (at 10 mg/kg or 20 mg/kg). Larvae inoculated with *E. coli* 042 and treated with PBS were used as the negative control; while larvae that received two doses of PBS were the positive control. The experiment was repeated three independent assays.

### 4.10. Nuclear Magnetic Resonance Analysis

The chemical characterization of the most active fractions (BTFC and BTFD) was performed by nuclear magnetic resonance (NMR) analysis. 1D and 2D NMR data were acquired at 298 K in DMSO-d_6_ on a Bruker AVANCE III 400 NMR spectrometer operating at 9.4 T, observing ^1^H and ^13^C at 400 and 100 MHz, respectively. The NMR spectrometer was equipped with a 5 mm direct detection probe (BBO) with a *z*-gradient. One-bond (^1^H-^13^C HSQC) and long-range (^1^H-^13^C HMBC) NMR correlation experiments were optimized for average coupling constant ^1^*J*_(C,H)_ and ^LR^*J*_(C,H)_ of 140 and 8 Hz, respectively. All ^1^H and ^13^C NMR chemical shifts (δ) were given in ppm related to the TMS signal at 0.00 as an internal reference and the coupling constants (*J*) in Hz.

### 4.11. Statistical Analysis

The results were expressed as the mean ± standard deviation (SD). Statistical significance was determined by one-way ANOVA or two-way ANOVA followed by Tukey and Bonferroni tests. A *p*-value < 0.05 was considered statistically significant. Determination of EC_50_ (half maximal effective concentration) was performed by linear regression. Correlations were assessed using the Pearson coefficient. The larvae survival assays were analyzed using the Kaplan–Meier method to calculate survival fractions, and the log-rank test was used to compare survival curves.

## 5. Conclusions

In this study, the use of in vitro antioxidant assays allowed the selection of fractions from a methanolic extract with a high activity and low toxicity. The fractions (BTFC and BTFD) were able to extend the lifespan of *T. molitor* larvae submitted to stress induced by heat-killed *E. coli* significantly. The therapeutic treatment with these fractions had also positive effects on the infection induced by the pathogenic strain *E. coli* 042 (EAEC). ^1^H NMR data indicated the presence of aliphatic (terpenes, fatty acids, carbohydrates) and aromatic compounds (phenolic compounds). These findings suggested that products derived from *B. tetraphylla* leaves are a promising candidate for the development of antioxidant agents able to treat the oxidative-related dysfunctions.

## Figures and Tables

**Figure 1 pharmaceuticals-13-00046-f001:**
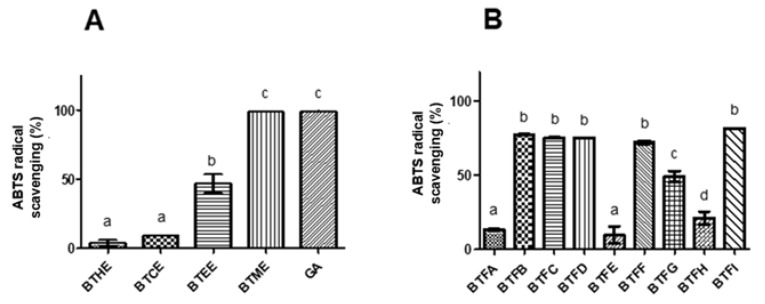
Comparative evaluation of antioxidant activity by the ABTS assay of the crude extracts (**A**) and fractions (**B**) of *Buchenavia tetraphylla* leaves. BTHE: hexane extract; BTCE: chloroform extract; BTEE: ethyl acetate extract; BTME: methanolic extract; BTFA: Fraction A; BTFB: Fraction B; BTFC: Fraction C; BTFD: Fraction D; BTFE: Fraction E; BTFF: Fraction F; BTFG: Fraction G; BTFH: Fraction H; BTFI: Fraction I. In each graph, values with significant differences (*p* < 0.05) are indicated by different superscript letters (^a, b, c^). The results are expressed as the mean ± standard deviation calculated from three independent assays performed in triplicate (*n = 3*).

**Figure 2 pharmaceuticals-13-00046-f002:**
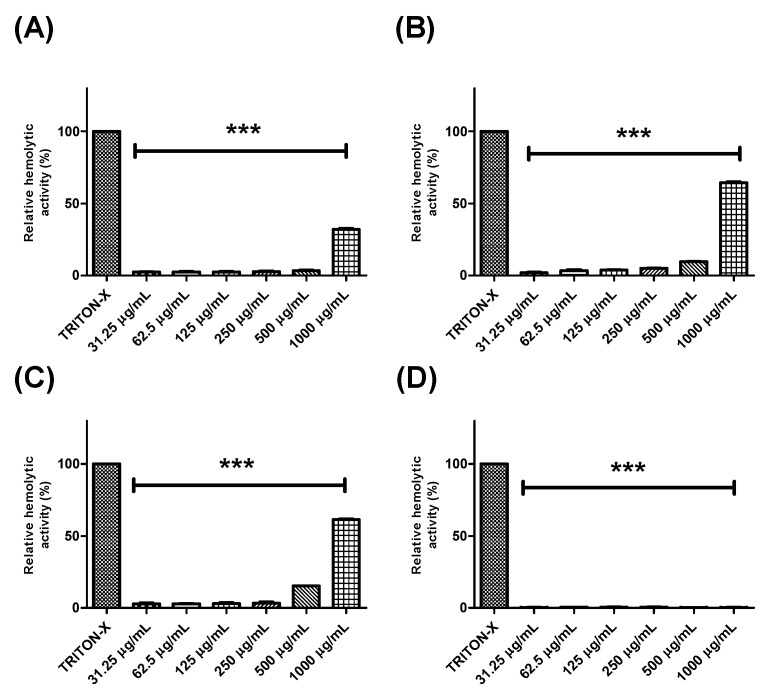
Hemolytic activity of the crude extracts of *Buchenavia tetraphylla* leaves. (**A**) BTHE (hexane extract); **(B**) BTCE (chloroform extract); (**C**) BTEE (ethyl acetate extract); (**D**) BTME (methanolic extract). *** Significant differences in relation to triton-X (*p* < 0.0001). The results are expressed as the mean ± standard deviation calculated from three independent assays performed in quadruplicate (*n = 4*).

**Figure 3 pharmaceuticals-13-00046-f003:**
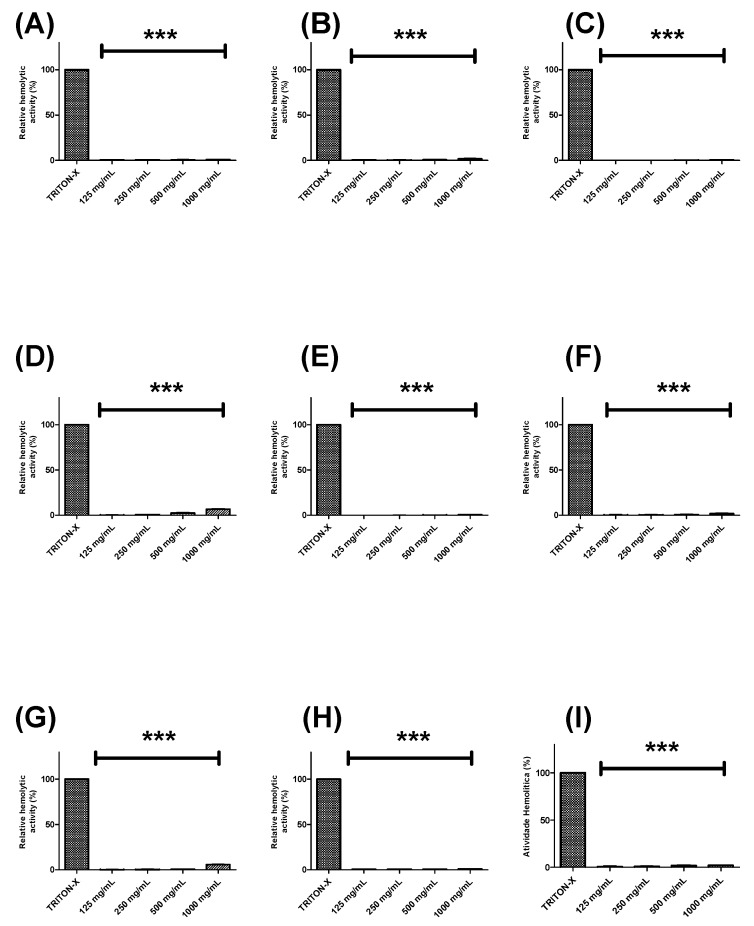
Hemolytic activity of the fractions obtained from the methanolic extract of *Buchenavia tetraphylla* leaves. (**A**) BTFA (Fraction A); (**B**) BTFB (Fraction B); (**C**) BTFC (Fraction C); (**D**) BTFD (Fraction D); (**E**) BTFE (Fraction E); (**F**) BTFF (Fraction F); (**G**) BTFG (Fraction G); (**H**) BTFH (Fraction H); (**I**) BTFI (Fraction I). *** Significant differences in relation to triton-X (*p* < 0.0001). The results are expressed as the mean ± standard deviation calculated from three independent assays performed in quadruplicate (*n = 4*).

**Figure 4 pharmaceuticals-13-00046-f004:**
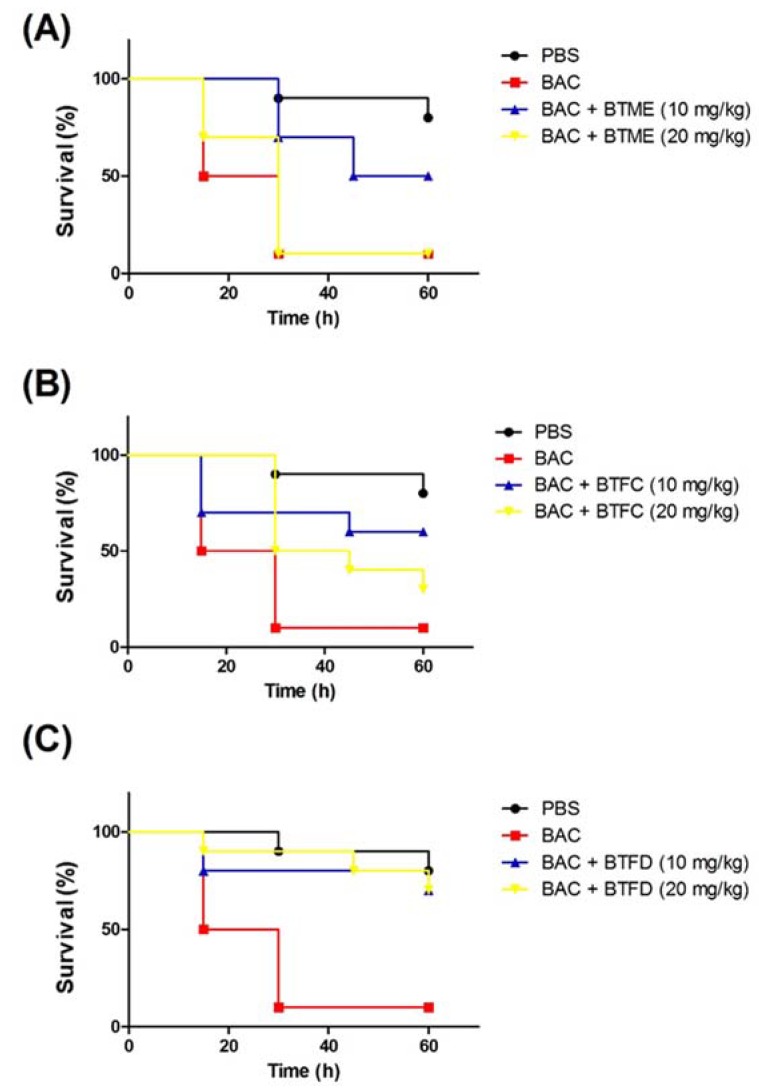
Effects of methanolic extract of *Buchenavia tetraphylla* leaves (BTME) and its fractions (BTFC and BTFD) on the survival of *Tenebrio molitor* larvae challenged with heat-killed *Escherichia coli* OP50. (**A**) Effects of BTME on the survival of *T. molitor* larvae challenged with heat-killed *E. coli* OP50; (**B**) effects of BTFC on the survival of *T. molitor* larvae challenged with heat-killed *E. coli* OP50; (**C**) effects of BTFD on the survival of *T. molitor* larvae challenged with heat-killed *E. coli* OP50. The larvae (*n* = 10/group) were pre-treated with each sample (at 10 mg/kg or 20 mg/kg) 12 h prior to inoculation of heat-killed bacteria. Larvae treated with phosphate-saline buffer (PBS) or *E. coli* OP50 (BAC) were used as negative or positive controls, respectively. In this set of assays, larvae survival was recorded each 12 h. The experiment was repeated three times.

**Figure 5 pharmaceuticals-13-00046-f005:**
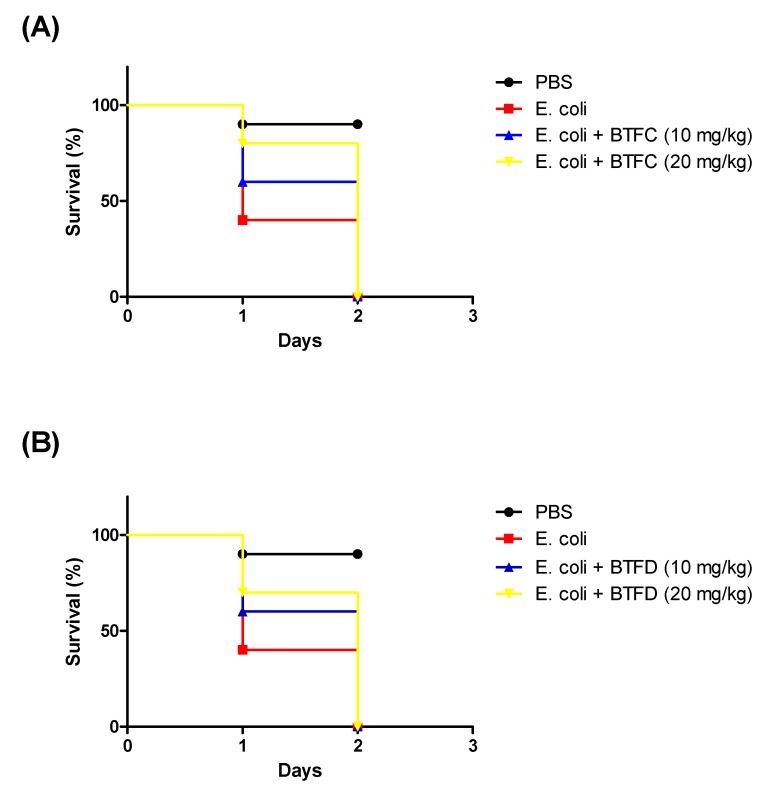
Effects of fractions from methanolic extract of *Buchenavia tetraphylla* leaves (BTFC and BTFD) on the survival of *Tenebrio molitor* larvae challenged with *Escherichia coli* 042. (**A**) Effects of BTFC on the survival of *Tenebrio molitor* larvae challenged with *E. coli* 042. (**B**) Effects of BTFD on the survival of *Tenebrio molitor* larvae challenged with *E. coli* 042. The larvae *(n = 10/group)* received a lethal dose of EAEC 042 and after 2 h were treated with fraction BTFC and BTFD (at 10 mg/kg or 20 mg/kg). Larvae treated with phosphate-saline buffer (PBS) or EAEC 042 were used as negative or positive controls, respectively. In this set of assays, larvae survival was recorded each 24 h. The experiment was repeated three times.

**Figure 6 pharmaceuticals-13-00046-f006:**
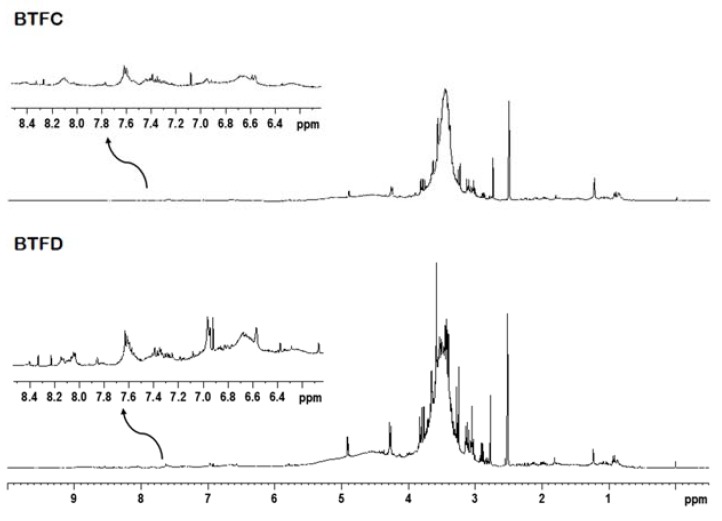
Representative ^1^H NMR spectrum of the active fractions (BTFC and BTFD) obtained from the methanolic extract of *Buchenavia tetraphylla* leaves.

**Table 1 pharmaceuticals-13-00046-t001:** Comparative analysis of total phenolic compounds, flavonoid content, and DPPH radical scavenging of the crude extracts from the leaves of *Buchenavia tetraphylla*.

	BTHE	BTCE	BTEE	BTME	Trolox
**Yield (%)**	5.30	14.48	10.45	14.44	-
**Phenolic compounds content (mg GAE/mg)**	9.45 ± 1.29 ^a^	26.53 ± 0.50 ^b^	116.65 ± 10.26 ^c^	123.03 ± 1.23 ^c^	-
**Flavonoid content (mg QE/mg)**	10.03 ± 0.14 ^a^	14.28 ± 0.48 ^b^	24.92 ± 0.45 ^c^	108.89 ± 0.06 ^d^	-
**DPPH (EC_50_ μg/mL)**	6826.45	3779.98	562.75	79.04	44.10

Legend: BTHE: hexane extract; BTCE: chloroform extract; BTEE: ethyl acetate extract; BTME: methanolic extract. In each row, the values with significant differences (*p* < 0.05) are indicated by different superscript letters (^a, b, c, d^). The results are expressed as the mean ± standard deviation calculated from three independent assays performed in triplicate (*n = 3*).

**Table 2 pharmaceuticals-13-00046-t002:** Comparative analysis of total phenolic compounds and flavonoid content and DPPH radical scavenging of the crude extracts from leaves of *Buchenavia tetraphylla*.

	BTFA	BTFB	BTFC	BTFD	BTFE	BTFF	BTFG	BTFH	BTFI
**Yields (%)**	0.10	0.29	1.22	2.53	2.04	0.96	0.46	0.56	0.98
**Phenolic contents (μg GAE/μg)**	49.44 ± 1.86 ^a^	107.20 ± 7.23 ^b^	155.67 ± 3.40 ^c^	168.98 ± 1.81 ^c^	49.80 ± 6.52 ^a^	72.83 ± 1.13 ^d^	127.62 ± 15.60 ^e^	55.61 ± 3.01 ^a^	110.10 ± 0.62 ^b^
**Flavonoid contents (μg QE/μg)**	16.80 ± 1.76 ^a^	12.65 ± 0.76 ^a^	68.27 ± 2.35 ^b^	56.01 ± 4.52 ^c^	14.01 ± 1.72 ^a^	4.28 ± 0.11 ^d^	45.27 ± 3.37 ^e^	0.86 ± 0.21 ^d^	39.29 ± 2.36 ^e^
**DPPH (EC_50_ μg/mL)**	2480.22	562.75	50.41	237.76	4132.98	2355.09	294.38	2578.18	376.25

**Legend:** BTFA-BTFI: Fractions obtained from *Buchenavia tetraphylla* methanolic extract. In each row, the values with significant differences (*p* < 0.05) are indicated by different superscript letters (^a, b, c, d, e^). The results are expressed as the mean ± standard deviation calculated from three independent assays performed in triplicate (*n = 3*).

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
