# Peer review of "Products Derived from Buchenavia tetraphylla Leaves Have In Vitro Antioxidant Activity and Protect Tenebrio molitor Larvae against Escherichia coli-Induced Injury"

_pharmaceuticals, 2020, doi:10.3390/ph13030046_

Round 1

Reviewer 1 Report

This work examines the antioxidant and antimicrobial effect of extracts obtained from B. tetraphylla leaves. While the premise of this study is interesting this reviewer raises several concerns regarding certain aspects of this study.

Several times throughout the results sections the authors choose to disseminate the data as a list of values, which has no real meaning or significance. These values need to be confined to the tables only. Regarding the tables - the legends need to be below the table. Not above and below as done within the manuscript. Abbreviations are used throughout the manuscript without being defined. Figure 1 has no control. These data are irrelevant without controls. Figure 2 there appears to be little to no effect on the hemolytic effect at several concentrations of these extracts compared to the 100% seen for the control. If this is the case then how can the stats say the significance is only <0.05? This doesn't seem to make sense. Figure 3 suffers from the same issues as in Figure 2. Also, it appears that every fraction has very limited hemolytic activity. One would assume that the fractions would be different based on the retention time in the column. This needs explaining.  It appears these fractions only protect against infection of the OP50 strain. There's no explanation or discussion for this in the manuscript which is curious given the interesting result. No n values are reported at all throughout the manuscript. How many times were these experiments repeated? These need to be reported and are critical information for an accurate review. What do the error bars represent? SEM, SD, etc. What statistical tests were used to achieve these p values?

Author Response

Dear reviewer, we are grateful for your contributions. We provide bellow a point-to-point discussion of each suggestion performed by you. The alterations made in the manuscript are highlighted in yellow.

Several times throughout the results sections the authors choose to disseminate the data as a list of values, which has no real meaning or significance. These values need to be confined to the tables only.

  • We have edited the text in order to improve the understanding of the text. The values are showed in the table, while the main findings are presented in the text.

Regarding the tables - the legends need to be below the table. Not above and below as done within the manuscript.

  • We have corrected this.

Abbreviations are used throughout the manuscript without being defined.

  • We have corrected this.

Figure 1 has no control.  These data are irrelevant without controls.

  • We have added to control (gallic acid).

Figure 2 there appears to be little to no effect on the hemolytic effect at several concentrations of these extracts compared to the 100% seen for the control. If this is the case then how can the stats say the significance is only <0.05? This doesn't seem to make sense. Figure 3 suffers from the same issues as in Figure 2.Also, it appears that every fraction has very limited hemolytic activity. 

  • We thank you so much for this observation and now we have changed for <0.001

It appears these fractions only protect against infection of the OP50 strain. There's no explanation or discussion for this in the manuscript which is curious given the interesting result.

  • We have added in the text our hypothesis that the protective effects of these fractions may be related with their antioxidant properties, since they did not display antimicrobial action toward coli in vitro. This explanation is located in the discussion section,

No n values are reported at all throughout the manuscript. How many times were these experiments repeated? These need to be reported and are critical information for an accurate review. What do the error bars represent? SEM, SD, etc. What statistical tests were used to achieve these p values? 

  • We have added the replications at the "statistic sections".

Reviewer 2 Report

The subject of manuscript is interesting and it matches to the scope of the special issue. Manuscript is also clearly written and well organized. However, I have some questions/suggestions for Authors (the detailed list is below).

- Move “(juices, teas, extracts, infusions)” from line 82 to line 81: “(…) products derived from plants (juices, teas, extracts, infusions)…”

- 2.1 Section : Avoid to repeat the same data in Tables and in text. For exemplary, lines 104-107 are repetition of data in Table 1 (the same comment for 2.2. section).

- Line 103: TPC is considered as an antioxidant assay (not quantitative test); thus the statement: “The methanolic extract presented higher concentrations of…” is uncorrect. Moreover: “higher” than what?

- Table 1: Is it the unit correct? µg GAE/QE per µg? FC 10-108   µg QE/µg? Lack of unit for IC50 (the same comment for Table 2)

-Line 108: It should be: “was found between the”

- The headlines for sections 2.3-2.5 it should not contain the conclusion.

- Fig. 4 – explain the shortcuts PBS, BAC in figure legend

- 2.6. Section. In my opinion the section should be removed because chemical characterization did not carried out. MNR showed only general profile. Moreover, it is very strange that in methanolic extract obtained after subsequent extraction of non- and lees polar solvents contained lipophilic compounds and only low amount of phenolic compounds (especially that FC values for methanolic extract was high).

- 4.2. Section: More detail is needed for fractionation procedure

- There are some editorial errors, e.g., different fonts (e.g. in Introduction), line 111: “legenda”, line 285: “obtention”, unnecessary capital letters, e.g. line 305 etc

The results are interesting but there is only preliminary study and I hope that Authors will continue their work in future.

Author Response

Dear reviewer, we are grateful for your contributions. We provide bellow a point-to-point discussion of each suggestion performed by you. The alterations made in the manuscript are highlighted in yellow.

The subject of manuscript is interesting and it matches to the scope of the special issue. Manuscript is also clearly written and well organized. However, I have some questions/suggestions for Authors (the detailed list is below).

- Move “(juices, teas, extracts, infusions)” from line 82 to line 81: “(…) products derived from plants (juices, teas, extracts, infusions)…”

- Thank you for this correction. We have performed it.

- 2.1 Section : Avoid to repeat the same data in Tables and in text. For exemplary, lines 104-107 are repetition of data in Table 1 (the same comment for 2.2. section).

- We have corrected this issue, thank you.

- Line 103: TPC is considered as an antioxidant assay (not quantitative test); thus the statement: “The methanolic extract presented higher concentrations of…” is uncorrect. Moreover: “higher” than what?

- We agree with the reviewer that TPC is also an antioxidant assay, however it is also considered a quantitative test since we express it as gallic acid equivalents.

- Table 1: Is it the unit correct? µg GAE/QE per µg? FC 10-108   µg QE/µg? Lack of unit for IC50 (the same comment for Table 2)

- Thank you for this observation, we have corrected for mg GAE/QE per mg as described in the

-Line 108: It should be: “was found between the”

- We have corrected this issue, thank you.

- The headlines for sections 2.3-2.5 it should not contain the conclusion.

- The use of this type of section description is usual in papers dealing with infection model.

- Fig. 4 – explain the shortcuts PBS, BAC in figure legend

- We have corrected this issue, thank you.

- 2.6. Section. In my opinion the section should be removed because chemical characterization did not carried out. MNR showed only general profile. Moreover, it is very strange that in methanolic extract obtained after subsequent extraction of non- and lees polar solvents contained lipophilic compounds and only low amount of phenolic compounds (especially that FC values for methanolic extract was high).

- 4.2. Section: More detail is needed for fractionation procedure

- We have added more details.

- There are some editorial errors, e.g., different fonts (e.g. in Introduction), line 111: “legenda”, line 285: “obtention” unnecessary capital letters, e.g. line 305 etc

- Thank you for your contributions, we have reviewed and corrected these issues throughout the text.

The results are interesting but there is only preliminary study and I hope that Authors will continue their work in future.

Reviewer 3 Report

It is opinion of the reviewer that this paper before acceptance needs several corrections. My individual comments are listed below.

43- It should be “… Sephadex LH-20 column chromatography”.

The yield of extraction by individual solvents should be reported.

69 – Glutathione is not a protein.

73 – The mechanisms of the antioxidant action should described in a better way.

Bioactive compounds from B. tetraphylla should reported in the Introduction section.

104 and other places – Results > 100 should be reported without any digitals after decimal point, 10 <result < 100 – with one digital, results < 10 with two digitals.

Table 1 – The unit of IC50?

112 & 153 – It should be “row” instead of ‘line”.

Table 2 – Remove from the footnote “BTHE … BTME”.

The antioxidant activity of the compounds classified using NMR should be discussed based on the literature.

291 – The column size?

293 – Type of the TLC plates? Mobile phase used?

301 – It should be “gallic acid equivalent”.

305 – It should be “aluminum chloride:.

307 – It should be “quercetin equivalent”.

311 – It should be “2,2-diphenyl …”.

318 – It should be “radical cation”.

315 & 323 – It should be “scavenging” instead of “inhibition” – free radical is scavenged not inhibited. I suggest to use a term of EC50 (Half maximal effective concentration) instead of IC50.

453, 454, 465, 483, 489, 493, 495, 500, 504, 508, 516, 523, 526, 527, 530m 537, 541, 546, 550 – Latin names must be in italic.

554, 557, 560 – The journal title abbreviation is needed.

563 – The reference is incomplete.

Author Response

Dear reviewer, we are grateful for your contributions. We provide bellow a point-to-point discussion of each suggestion performed by you. The alterations made in the manuscript are highlighted in yellow.

It is opinion of the reviewer that this paper before acceptance needs several corrections. My individual comments are listed below.

43- It should be “… Sephadex LH-20 column chromatography”.

  • We have corrected this throughout the text

The yield of extraction by individual solvents should be reported.

  • We added these values in each table

69 – Glutathione is not a protein.

  • We have corrected this by change it for ‘Glutathione reductase”

73 – The mechanisms of the antioxidant action should described in a better way.

  • The mechanisms are explained in the next sentences (lines 74-77).

Bioactive compounds from B. tetraphylla should reported in the Introduction section.

  • We have included the description of the compounds already isolated in this plant (see lines 91-93.

104 and other places – Results > 100 should be reported without any digitals after decimal point, 10 <result < 100 – with one digital, results < 10 with two digitals. Table 1 – The unit of IC50?

  • We decided to use two digits for all values. We have added the unit of IC50 in the table.

112 & 153 – It should be “row” instead of ‘line”.

  • We performed these alterations.

Table 2 – Remove from the footnote “BTHE … BTME”.

  • We performed these alterations.

The antioxidant activity of the compounds classified using NMR should be discussed based on the literature.

291 – The column size? 293 – Type of the TLC plates? Mobile phase used?

  • We have added these data.

301 – It should be “gallic acid equivalent”. 305 – It should be “aluminum chloride”. 307 – It should be “quercetin equivalent”. 311 – It should be “2,2-diphenyl …”. 318 – It should be “radical cation”.

  • We performed these alterations.

315 & 323 – It should be “scavenging” instead of “inhibition” – free radical is scavenged not inhibited. I suggest to use a term of EC50 (Half maximal effective concentration) instead of IC50.

  • We changed these terms.

453, 454, 465, 483, 489, 493, 495, 500, 504, 508, 516, 523, 526, 527, 530m 537, 541, 546, 550 – Latin names must be in italic. 554, 557, 560 – The journal title abbreviation is needed. 563 – The reference is incomplete

  • We performed these alterations.

Round 2

Reviewer 1 Report

While this is an improvement of the original manuscript I still have significant issues.

I am perplexed and extremely concerned about the author's reluctance to provide accurate n values for the experiments they conducted. Why is this so difficult? It is a common practice for every journal and without these values the data is irrelevant. Without these accurate n values, how can the reader derive any meaning from the graphs, stats etc. 

Also, the way statistics are being reported by the authors remains a little baffling. In figure 1 a blanket statement is used to suggest statistical significance is determined as being <0.05, yet the authors then decided to use letters to differentiate within this broad statement without providing a numerical value for each of the letters.

I would prefer to see p values for each statistical comparisons made, irrespective of significance. However, this is a new remit for scientific journals so I can't expect this to occur within this manuscript. But the authors need to clean this up because right now it is very vague.

Line 126-127 -  'In each graph values with significant differences (p < 0.05) are indicated by different superscript letters'. What are the values for a, b, c, d etc.? Currently, it makes no sense to differentiate when you provide a broad statement like the above.

Line 373 - This is far too vague. You need to report an accurate n value for each experiment and this needs to be included in the figure legend. Stating you repeated the experiments 'at least 3 times' in the methods section is not sufficient. 

Author Response

Dear reviewer, 

Thank you for your comments.

We have now provided the specific number of replicates for all assay, as well as in each figure legend.

Reviewer 3 Report

The authors corrected this paper properly taken under considerations all my comments. Therefore, I can accept it now.

Author Response

Dear reviewer, thank you so much for accept our manuscript.

Round 3

Reviewer 1 Report

Fine